# The Mode of Cytokinin Functions Assisting Plant Adaptations to Osmotic Stresses

**DOI:** 10.3390/plants8120542

**Published:** 2019-11-26

**Authors:** Ranjit Singh Gujjar, Kanyaratt Supaibulwatana

**Affiliations:** 1Department of Biotechnology, Faculty of Science, Mahidol University, Bangkok 10400, Thailand; ranjit.gujjar@icar.gov.in; 2Division of Crop Improvement, Indian Institute of Sugarcane Research, Lucknow 226002, India

**Keywords:** cytokinin (CK), osmotic stress, delayed senescence, isopentenyl transferase (IPT), oxidative stress, growth and yield, abscisic acid (ABA) antagonism

## Abstract

Plants respond to abiotic stresses by activating a specific genetic program that supports survival by developing robust adaptive mechanisms. This leads to accelerated senescence and reduced growth, resulting in negative agro-economic impacts on crop productivity. Cytokinins (CKs) customarily regulate various biological processes in plants, including growth and development. In recent years, cytokinins have been implicated in adaptations to osmotic stresses with improved plant growth and yield. Endogenous CK content under osmotic stresses can be enhanced either by transforming plants with a bacterial isopentenyl transferase (IPT) gene under the control of a stress inducible promoter or by exogenous application of synthetic CKs. CKs counteract osmotic stress-induced premature senescence by redistributing soluble sugars and inhibiting the expression of senescence-associated genes. Elevated CK contents under osmotic stress antagonize abscisic acid (ABA) signaling and ABA mediated responses, delay leaf senescence, reduce reactive oxygen species (ROS) damage and lipid peroxidation, improve plant growth, and ameliorate osmotic stress adaptability in plants.

## 1. Introduction

Osmotic stresses in plants are caused by drought, salinity and high temperature. Drought is a major global problem that impacts 1–3% of all land. This number is predicted to increase to 30% by 2090 [1]. Soil salinity is also a worldwide problem that impacts 397 million hectares of land [2]. Osmotic stress tolerance is a complex process involving a myriad of signaling pathways and results in either upregulation or downregulation of innumerable genes. These genes may either be regulatory (i.e., transcription factors that may further induce some other genes) or functional (i.e., directly involved in the process of stress tolerance). Strategies used to create osmotic stress tolerant transgenic plants involve overexpression of such regulatory or functional genes. The ultimate aim of plants under osmotic stress is to survive with minimal metabolic processes, which results in sluggish growth [3]. Recent findings posit the role of cytokinins (CKs) in mediating cellular responses to drought acclimation [4]. Exogenous application of cytokinins improves plant soluble sugar contents which in turn act as osmolytes to help plants tolerate osmotic stresses [5]. Moreover, CKs play a crucial role in delaying senescence, reducing oxidative damage and upholding plant growth under osmotic stresses. Targeted control of CK metabolism is a powerful tool to develop drought-tolerant plants [6,7,8,9,10]. This review assesses functions of enhanced CK contents in improving osmotic stress adaptability of plants without compromising yield.

## 2. Cytokinin Metabolism and Signal Transduction during Osmotic Stress

CKs primarily control cell growth and differentiation in plants. In response to osmotic stresses, concentration and transport of CKs in plants decrease drastically, whereas abscisic acid (ABA) levels increase [4,11,12,13,14,15]. Reduced CK levels inhibit cell division and cell growth. On the other hand, enhanced concentrations of ABA and ethylene promote the closure of stomatal apertures, thereby reducing water loss through transpiration, and activate the senescence-related Cys protease (phcp1) that promotes senescence of old leaves [16,17,18]. Characteristically, plants survive with minimum resources under osmotic stress. Two major enzymes concomitant with CK metabolism in plants are isopentenyl transferase (IPT), which is involved in the synthesis of CKs and cytokinin oxidase (CKX) that degrades CKs [19,20,21]. Genome surveys of different crops revealed the presence of more than one gene encoding each of these enzymes. In Arabidopsis, there are 9 *IPT* (*AtIPT1* to *AtIPT9*) genes and 7 *CKX* (*AtCKX1* to *AtCKX7*) genes [22,23] while in soybean, 14 CK biosynthetic (*GmIPT*) and 17 CK degradative (*GmCKX*) genes have been identified [19]. The high number of abiotic stress-inducible cis-elements on the promoter of *CKX* genes causes their upregulation under water deficit stress resulting in decreased CK levels [19,24,25,26]. Conversely, *IPT* transcript levels remain almost stationary in response to osmotic stress [19].

CK signal transduction culminates in the regulation of CK responsive genes and has been studied in several crop species, most extensively in Arabidopsis and rice [14,27,28,29,30,31]. CK signal perceptions and transduction pathways consist of three major components as histidine kinase (HK) receptors, authentic histidine phosphotransferases (AHPs) and response regulators (RRs) [32,33]. Signal transduction follows a multistep His-Asp phosphorelay, which involves four consecutive phosphorylation events that alternate between histidine and aspartate residues [34]. HK receptors are predominantly localized on endoplasmic reticulum (ER) membranes, while plasma membrane is reported to have relatively fewer HK receptors [35]. HK receptors have a conserved CK-binding CHASE (cyclases/histidine kinases-associated sensing extracellular) domain, two transmembrane domains, a histidine kinase domain, and two receiver domains [36]. During the process of CK perception and transduction, CK binding to the CHASE domain activates the cytosolic HK domain, which then undergoes an auto-phosphorylation of its conserved His residue. The phosphate is then transferred to a conserved Asp residue within the receiver domain [33,37]. The phosphate group is further transferred to downstream AHPs, which act as high-energy phosphodonors. AHPs have no catalytic activity and transfer the phosphate to an Asp residue within a type-B RR receiver domain [38]. Besides possessing a phosphate receiver domain, type-B RRs also have a large C-terminal extension that contains a Myb-like DNA-binding domain. DNA-binding motifs for type-B RRs have been identified upstream in many CK-regulated genes [39,40]. The process of CK signal transductions involves two checkpoints at the level of AHPs and type-B RRs for negative regulation. Pseudo-histidine phosphotransfer proteins (PHPs) mimic AHPs to accept phosphate from HK receiver domains but lack the histidine phosphorylation site to further transfer the phosphate group [37]. Plants also have type-A RRs that contain a phosphate receiver domain similar to type-B RRs; however, they lack a classic output domain to regulate transcription of CK genes [41]. Dephosphorylated type-A RR proteins are usually unstable and degraded by 26S proteasomes. Phosphorylation of type-A RRs leads to autoregulation of their transcription to ensure availability in response to CK signaling [42,43].

## 3. Cytokinins Antagonize ABA Signaling and ABA Arbitrated Adjustments during Osmotic Stress

CKs and ABA are unfriendly phytohormones, but both are equally vital for plant developmental processes. ABA promotes stress-induced signaling that enables plants to adjust under unfavorable environmental conditions [44,45]. ABA signaling stimulates CKX activity during osmotic stress and reduces CK concentration [19,26]. By contrast, elevated concentrations of CKs during osmotic stress conditions, either through exogenous application or *IPT* overexpression, counteract ABA arbitrated events [4,13]. Increased levels of CKs are known to affect both biosynthesis and sensitivity of ABA in plants under osmotic stress conditions. Key components of ABA signaling are SnRK2 (sucrose non-fermenting-1-related protein kinase 2) protein kinases, which phosphorylate downstream targets and trigger ABA-induced responses in plants. When ABA concentration is low, SnRK2 protein kinase activity is inhibited by PP2C (protein phosphatase type-2C) phosphatases. When ABA concentration increases during stress/exogenous application, ABA binds to its receptors, namely PYR/PYL/RCARs (pyrabactin resistance/ pyrabactin-like/ regulatory components of the ABA receptor), which in turn bind to PP2Cs and inactivate them. Consequently, PP2Cs are dissociated from SnRK2s resulting in activation of SnRK2s to initiate ABA-induced responses [46,47,48]. Crosstalk between signaling components of both ABA and CK reveals antagonism between the two phytohormones (Figure 1). Under osmotic stress conditions, when ABA concentration is high with prominent signaling in plants, SnRK2s phosphorylate and activate ARR5 (type-A response regulator 5). Phosphorylated AAR5 proteins auto-activate their transcription and competitively inhibit phosphate transfer to type-B RRs, thus hampering CK signaling. However, if CK concentration is high (in case of exogenous application/*IPT* overexpression) under osmotic stress, CK signaling components inhibit ABA signaling. Type-B RRs, which are positive regulators of CK signaling, inhibit SnRK2 activity [15].

Increase in CK concentration in *IPT* overexpressing tobacco plants resulted in suppression of ABA biosynthetic genes [49]. Likewise, overexpression of *IPT8* in Arabidopsis caused insensitivity in ABA treatments and prevented induction of *ABI1* and *ABI5* in seedlings [50]. Leaf fragments of *Commelina* and maize were assessed for the effects of CK and ABA treatments on stomatal behavior. When applied separately, ABA triggered stomatal closure in both *Commelina* and maize, while CKs (zeatin and kinetin at 1 to 100 m^−3^ mol/L) did not promote stomatal opening in either species. However, when applied simultaneously, CKs reversed ABA-induced closure of maize stomata but showed no effect on ABA-induced closure of *Commelina* stomata [51]. Under white light illumination, 10 μM ABA almost completely closed the stomata in isolated epidermal peels of Arabidopsis, and subsequent application of 10 μM BA (6-benzyladenine) reversed ABA-induced stomatal closure. Nevertheless, BA application had no effect on dark-induced stomatal closure, nor did it enhance stomatal opening under light conditions when applied separately. Results confirmed that CKs only revert stomatal closure induced by ABA and not under normal conditions [52]. Effects of exogenous application of benzyladenine (BA) and abscisic acid (ABA) were investigated separately as well as simultaneously on stomatal gas exchange in *Phaseolus vulgaris* L. leaves. When sprayed separately, 100 μM ABA decreased stomatal conductance, transpiration rate and net photosynthetic rate, while 10 μM BA had no significant effect on these parameters individually. By contrast, when applied simultaneously, application of 10 μM BA reversed the effects of 100 μM ABA on common bean leaves [53]. Furthermore, BA delayed the development of water deficit stress and increased photosynthetic rate in stressed leaves [53].

## 4. IPT Overexpression Influences Growth and Osmotic Stress Adaptability

Plants exhibit enhanced acclimation to osmotic stresses when endogenous CK content is high. The *IPT* gene catalyzes the rate-limiting step of cytokinin biosynthesis. Overexpressing *IPT* genes under control of senescence-induced pSAG12 or maturation and stress-induced pSARK promoters are the preferred approaches among researchers to manipulate CK levels [54,55]. The *IPT* gene from *Agrobacterium tumefaciens* has been frequently targeted for this purpose [56]. Increased CK contents through overexpression of the *IPT* gene, driven by senescence-activated promoter (*pSAG12::IPT*) in creeping bentgrass (*Agrostis stolonifera*), imparted enhanced tolerance to drought [8] and alleviated drought-induced damages to promote root growth [10]. Enhanced drought tolerance ability of *IPT* overexpression in *A*. *stolonifera* plants is attributed to accumulation of some essential metabolites, predominantly amino acids (proline, g-aminobutyric acid, alanine and glycine), carbohydrates (sucrose, fructose, maltose and ribose), and organic acids that are mainly involved in the citric acid cycle. These metabolites promote drought tolerance due to their well-known roles in osmotic adjustment, stress signaling and respiration for energy production [8]. In another experiment, *IPT* overexpression was investigated in *Agrostis stolonifera* plants under the control of SAG12 (*pSAG12::IPT*) and HSP18.2 (*pHSP18.2::IPT*) promoters. Transgenic lines showed higher CK contents with better drought adaptability and maintained a higher CK-to-ABA ratio. Furthermore, transgenic lines had better turf quality, photochemical efficiency, chlorophyll content, and leaf relative water content (RWC) under drought stress than NT (null transformant) plants [57].

Overexpression of the *IPT* gene, driven by SARK promoter (*pSARK::IPT*), augmented the synthesis of CK and contributed to enhanced osmotic stress tolerance in tobacco (*Nicotiana tabacum*) transgenic plants [58,59]. The transgenic plants displayed minimal reduction in biomass and seed yield (8–14%) under water deficit stress, compared to WT (wild type) plants, which showed a decline of 57% and 60% in biomass and seed yield, respectively. Improved yield of *IPT* overexpressing plants may be ascribed to a better photosynthetic rate and 2–3 times higher water use efficiency (WUE) compared to WT plants [58]. Synchronized expression of *IPT* (*pSARK::IPT*) significantly improved drought adaptability of peanut plants in both laboratory and field conditions. Transgenic peanut plants sustained higher photosynthetic rates, higher stomatal conductance, and higher transpiration than WT control plants under reduced irrigation conditions. Consequently, transgenic plants produced substantially higher yields than control plants in the field [7].

In rice plants, overexpression of *IPT* (*pSARK::IPT*) demonstrated greater tolerance to osmotic stress [60,61]. Enhanced stress adaptability of transgenic rice plants was facilitated by cytokinin-dependent harmonized regulation of carbon and nitrogen metabolism, which resulted in a healthier source to sink relationship under stress conditions [60,61]. Transgenic rice plants showed delayed response to stress, with significantly higher biomass and grain yield compared to WT plants [60]. A batch of *pSARK::IPT* transgenic cotton plants, grown in growth chamber condition, demonstrated enhanced tolerance to drought stress [9], which was accredited to delayed senescence of leaves and flowers. The *IPT* transgenic plants produced more root and shoot biomass, dropped fewer flowers, maintained higher chlorophyll contents, and higher photosynthetic rates under reduced irrigation conditions in comparison to the wild-type and segregated non-transgenic lines [9]. A gene expression study of *IPT* exposed a noteworthy shift in expression of hormone-related genes in transgenic plants. During water deficit stress, *pSARK::IPT* plants displayed increased transcription of brassinosteroid (BR)-related genes and repression of jasmonate (JA)-related genes [9,60]

Improved drought tolerance with delayed leaf senescence was displayed by *pSAG12::IPT* transgenic cassava plants. Detached leaves of the transgenic plants retained more chlorophyll compared to WT plants. Transgenic plants accumulated more trans-zeatin-type cytokinins with positive effects on photosynthesis, sugar allocation, and nitrogen partitioning [6]. Eggplants transformed with *pSAG12::IPT* also exhibited delayed senescence along with tolerance to drought and cold stresses. Compared to WT plants, transgenic eggplants displayed higher contents of chlorophyll, indole acetic acid (IAA), zeatin riboside, and gibberellic acid (GA) while ABA and malondialdehyde levels were low. Consequently, vegetative growth rates and yields of transgenic plants were relatively higher than those of WT plants [62]. In agreement with *IPT* gene insertions resulting in enhanced CK concentration and improved osmotic stress tolerance, CKX gene knockouts have also been tested for tolerance under salinity stress. Knockdown of *OsCKX2* in rice using the RNAi-based approach resulted in a significant increase in cytokinins under salt stress condition. *OsCKX2*-knockdown plants displayed improved vegetative growth, relative water content, and photosynthetic efficiency compared to wild types under salinity stress [63].

Generally, elevated CK levels under water deficit stress condition are associated with enhanced osmotic stress tolerance and better growth. Nevertheless, some contrasting observations were also reported where decreased biosynthesis of CK in transgenic plants resulted in improved root growth and drought tolerance. Transgenic Arabidopsis and tobacco plants, with enhanced root-specific degradation of CK, exhibited larger root systems compared to WT plants with no effect on growth and development of shoots. Moreover, transgenic plants displayed higher survival rates after severe drought treatment [25]. In another investigation, barley (*Hordeum vulgare*) was transformed using the *CKX1* gene from *Arabidopsis thaliana* (*AtCKX1*) under the control of mild root-specific b-glucosidase promoter. The transgenic plants were found to have low CK contents with better root systems and maintained high water contents under severe water deficit stress compared to WT plants [64].

## 5. Cytokinin Mediated Drought Acclimation is Primarily due to Delayed Senescence

Senescence is a natural physiological aging phenomenon in plants. The process involves the degradation of macromolecules, which mobilize nutrients from senescing tissues to sink tissues in order to sustain growth and development. Besides natural aging factors, senescence may also be triggered by various biotic and abiotic stresses [65,66]. Leaf senescence, as the final stage of development, follows a synchronized order of events involving loss of chlorophyll with subsequent reduction in photosynthesis, degradation of macromolecules, relocation of nutrients, dismantling of cellular components and, finally, cell death [67,68,69]. Senescence of plant leaves and flowers is achieved by the coordinated action of numerous senescence-associated genes (SAGs) with cysteine proteases as key components [70,71,72].

In the context of phytohormones, senescence is complemented by a decline in leaf CK content. An upsurge in the endogenous concentration or exogenous application of CKs results in nutrient mobilization and delays senescence [8,10,73]. Extracellular invertase and hexose transporters, responsible for apoplasmic phloem unloading, are co-induced by elevated levels of CKs, which instigate delay of senescence via an effect on source-sink relations [74]. Enhanced expression of the *IPT* gene from *Agrobacterium tumefaciens*, under the control of senescence-associated gene promoter (SAG12) is a widely used approach to delay senescence [6,8,75]. This *IPT* overexpression in plants with delayed senescence is useful for studying interactions of signaling mechanisms pertaining to CK based stress tolerance [6,62].

Environmental stresses, particularly drought, are responsible for premature leaf and flower senescence in plants by inducing synthesis of different types of cysteine proteases [76,77,78]. Delaying senescence through elevated levels of cytokinin during such environmental stresses has been proved beneficial for plants in terms of drought adaptation. Overexpression of the *IPT* gene under control of maturation and stress-induced promoter (*pSARK::IPT*) was reported to delay drought-induced senescence and thereby create drought tolerance in tobacco [58,59], peanut [7], rice [60,61] and cotton [9]. Likewise, *IPT* transformations under control of senescence-associated gene promoter (*pSAG12::IPT*) also proved advantageous in delaying senescence and imparting drought tolerance in cassava [6], *Agrostis stolonifera* [8,57] and brinjal [62].

Effects of high CK content on ethylene synthesis and sensitivity and ABA accumulation were examined in petunia plants transformed with *IPT* under control of pSAG12 promoter (*pSAG12::IPT*). Floral senescence in transgenic lines was delayed by 6 to 10 days compared to WT flowers. Endogenous ethylene biosynthesis was induced by pollination in both transgenic and WT flowers but biosynthesis was postponed in *IPT* transgenic flowers. Moreover, flowers from *IPT* transgenic plants were relatively less sensitive to exogenous ethylene and required longer treatment times to induce endogenous ethylene production, corolla senescence and upregulation of the senescence-related Cys protease phcp1. ABA accumulation was relatively less in flowers of *IPT* transgenic plants [79]. Cassava plants transformed with *pSAG12::IPT* exhibited delayed senescence under drought stress and retained relatively high levels of chlorophyll compared to WT plants. Induced expression of *IPT* also had positive effects on photosynthesis, sugar allocation and nitrogen partitioning. Furthermore, the transgenic lines showed significant drought tolerance as indicated by stay-green capacity after drought stress treatments [6].

## 6. Cytokinins Uphold Plant Growth during Abiotic Stresses

One of the prime objectives of plant biologists is to improve plant performances under less favorable environmental conditions. Approaches frequently used to counter drought stress are largely based on the overexpression of either regulatory or functional genes that are upregulated during stress conditions [80,81,82,83]. Transgenic plants created by altering the expression of such drought responsive genes are capable of withstanding drought up to a certain extent but exhibit a considerable drop in yield [84,85]. This loss of yield under drought stress is attributed to reduced CK biosynthesis followed by stress adaptive plant mechanisms including remobilization of nutrients triggering senescence, closure of stomata leading to reduced transpiration and gaseous exchange, and reduced photosynthetic rate [26] (Figure 2). Ultimately, plants follow the strategy of “survive with minimum” during stress conditions.

The novelty of CK mediated drought acclimation during the past decade has revolutionized the research pertaining to stress tolerance. Transgenic plants generated by overexpression of the *IPT* gene delivered better yield and tolerated drought for an extended length of time. Improved yield of *IPT* transgenic plants under limited water conditions may be ascribed to higher CK levels that counteract leaf senescence by mobilizing the remobilized nutrients [6,10,62], improving photosynthetic efficiency [7,58,92], interrupting drought-induced ABA signaling [15,50,93], and eventually stopping all those events that guide the plant to “survive with minimum” (Figure 3).

Leaf discs of *pSAG12::IPT* modified gerbera plants, incubated in 40% (w/v) polyethylene glycol (PEG) for 20 h under continuous light [130 μmol/(m^2^·s)], retained relatively higher contents of chlorophyll, carotenoids, and soluble proteins compared to control plants [96]. Tomato roots, transiently expressing the *IPT* gene (*pHSP70::IPT*), exhibited a 2–3-fold increase in root CK concentration and improved plant growth and yield under salinity stress (100 mM NaCl for 22 days). Enhanced CK concentrations in transgenic tomato plants delayed stomatal closure and leaf senescence and virtually doubled shoot growth compared to WT plants. Furthermore, ABA and toxic Na^+^ ion concentrations decreased by 20–40% and 30% respectively with concomitant increases in the essential K^+^ ion by 20% in mature leaves [93]. In another experiment, WT shoots were grafted onto a constitutive *IPT* expressing rootstock (*p35S::IPT*); plant yield was enhanced by 30% compared to WT under salinity stress [93]. High yield and superior growth rate of *IPT* overexpressing plants may also be substantiated by improved N-use efficiency. Transgenic tobacco plants (*pSARK::IPT*) with increased CK biosynthesis, maintained adequate biomass and growth rates under limited N conditions compared to WT plants [97]. Higher CK biosynthesis in *pSAG12::IPT* modified rice plants resulted in vigorous growth even after a long drought period that killed the control plants. The transgenic plants displayed improved photosynthetic activity and maintained high water contents during drought [58]. Similar modifications (*pSAG12::IPT*) in transgenic peanut plants also resulted in higher photosynthetic rates with improved stomatal conductance and transpiration, compared to WT control plants under limited irrigation conditions. All these CK mediated changes in transgenic peanut plants resulted in significantly higher yields than wild-type control plants in the field [7]. Experiments involving CK-deficient mutants also revealed reduced growth and yield. Loss-of-function mutants of CK receptors *AHK2*, *AHK3* and *CRE1*/*AHK4* in *Arabidopsis* displayed rapid seed germination but the leaves of mutants formed fewer cells and had reduced chlorophyll content [98]. Arabidopsis CK-deficient *ipt1, 3, 5* and *7* mutants showed better salt and drought tolerance but had reduced yield compared to WT [26].

Elevated concentrations of CK under drought stress counteract drought-induced signaling and facilitate plants to act ordinarily [4,99]. Hence, plants with high CK concentration are able to maintain normal levels of leaf water content, photosynthetic rate and stomatal conductance even under stress conditions. All these factors cooperatively lead to better growth and development of plants under stress. Besides *IPT* modulations, exogenous applications of synthetic cytokinins have also proved equally beneficial in improving plant growth under stress environments. Foliar spray of CPPU enhanced salt tolerance in rice by maintaining rates of photosynthesis, soluble sugars and free proline concentration under salinity stress [92]. Exogenous application of 100 μM CK onto creeping bentgrass improved turf quality and delayed leaf wilting under drought stress and elevated N conditions [94]. In a recent study on wheat, exogenous application of the optimized dose of 10 mg L^−1^ BAP significantly increased membrane stability index (MSI), photosynthetic pigment contents, chlorophyll stability index, and other growth parameters under drought and high temperature stress conditions [100].

## 7. Cytokinins Moderate ROS Levels during Osmotic Stresses

Reactive oxygen species (ROS) are generated in chloroplasts and mitochondrial electron transport chains in response to both abiotic and biotic stresses [88]. Drought-induced oxidative damage can lead to lipid peroxidation, protein degradation and nucleotide damage, further inhibiting a wide range of plant cellular processes [90,91,101]. Major ROS scavenging enzymatic systems in plants are dehydroascorbate reductase (DHAR), ascorbate peroxidase (APX), guaiacol peroxidase (GPX), glutathione reductase (GR), superoxide dismutase (SOD), catalase (CAT), and peroxidase (POD) [102,103]. Increasing a plant’s CK contents either through exogenous application [104,105] or by overexpression of *IPT* [8,57,93,106,107] can mitigate oxidative stress and improve drought tolerance. CKs do this by positively modulating antioxidant enzymatic activities (i.e., POD, SOD and CAT) of plants, thereby aiding plant defenses to abiotic stresses [95,104,106,108,109,110].

Tobacco plants overexpressing the *IPT* gene under control of the promoter of a small subunit of rubisco (*pSSU::IPT*) exhibited delayed senescence and predominance of zeatin and zeatin riboside type CKs [95]. The transgenic plants demonstrated increased activities of antioxidant enzymes (CAT, GR, APX) compared to control plants [95,111]. Furthermore, electron microscopic investigation revealed relatively higher numbers of crystal-like pores in peroxisomes and abnormal interactions among organelles in transgenic tobacco plants [95]. In another report, *pSARK::IPT* transgenic tobacco plants grown under limited nitrogen (N) conditions demonstrated reduced oxidative damage and higher biomass compared to WT plants [97]. Gerbera plants, modified by the *pSAG12::IPT* chimeric gene and induced by 40% (w/v) polyethylene glycol (PEG) mediated osmotic stress for 20 h, revealed higher activities of SOD, CAT, APX, GPX and DHAR compared to control plants [96]. Additionally, transgenic gerbera plants showed reduced lipid peroxidation rate (measured by thiobarbituric acid reactive substance (TBARS)) under PEG-induced osmotic stress [96].

*pSAG12::IPT* modification in creeping bentgrass (*Agrostis stolonifera* L.) led to greater antioxidant enzyme activities of SOD, POD and CAT, with relatively lower lipid peroxidation in leaves under osmotic stress than NT plants [8]. In another experiment, creeping bentgrass was modified using the *pSAG12::IPT* gene cassette. Root physiological analysis of transgenic plants showed significantly lower contents of ROS (hydrogen peroxide and superoxide) and less lipid peroxidation compared to WT roots under drought stress. Enzymatic assays and transcript abundance analysis further revealed predominantly higher activities of SOD, POD, CAT and DHAR in roots of transgenic bentgrass under drought stress [10]. Antioxidant metabolism of drought-stressed creeping bentgrass was also studied under simultaneous effects of CK (0, 10 and 100 μM) and N (low = 2.5 and high = 7.5 kg N/ha every 15 days) applied exogenously. Plants with CK10 and CK100 treatments had lower O_2_^−^ and H_2_O_2_ concentration than control CK0 plants. The CK100 treatment boosted activities of SOD, APX, CAT and POD by 25%, 22%, 17%, and 24%, respectively compared to CK0 [94]. However, the activity of these antioxidant enzymes enhanced more significantly under high N condition relative to low N condition, in contrast to findings of Rubio-Wilhelmi [97], where *pSARK::IPT* transgenic tobacco plants showed less oxidative damage under limited N conditions. Nevertheless, enhancing the concentration of CK, either through exogenous application or endogenously via *IPT* modifications, always proved advantageous for the plant in terms of reducing oxidative damage.

Plenty of evidences suggest that drought induces senescence of leaves with concurrent enhancement in ROS production. Both senescence and ROS production are moderated by higher CK levels inside the plant. However, when plants are not under stress, contrasting results deny the correlation of senescence progression with ROS-triggered lipid peroxidation. Recently, leaves of *pSAG12::IPT* overexpressing tobacco plants were allowed to undergo natural senescence and lipid peroxidation rate was determined by GS-MS using end product malondialdehyde (MDA). Though leaves of *pSAG12::IPT* remained green due to delayed senescence, lipid peroxidation was much higher compared to WT leaves of the same age [112]. Results indicated that lipid peroxidation cannot be correlated with leaf senescence. Further, ROS generation is not always moderated by higher CK levels in plants; however, stress-induced enhancement of ROS is controlled by elevated CK levels as evidenced by earlier reports.

## 8. Cytokinin-Induced Transcriptomic and Proteomic Changes during Osmotic Stresse

A transcriptome study of *pSARK::IPT* modified tobacco plants, performed under prolonged water deficit conditions, revealed repression of carotenoid pathway genes which are implicated in ABA biosynthesis [49]. By contrast, higher transcript abundance of genes involved in brassinosteroid biosynthetic pathways was witnessed in transgenic plants. Furthermore, transgenic plants displayed significantly higher transcript levels of genes associated with PSI, PSII, cytochrome b6/f complex, NADH oxidoreductase and the adenosine triphosphate (ATP) complex. Differential transcript levels in *pSARK::IPT* tobacco plants were further complemented by assessing expression of corresponding proteins using Western Blot [49]. CK arbitrated transcriptome changes, as observed by overexpression of *IPT* (*pSAG12::IPT*) in creeping bentgrass under drought stress, exposed differential expression of 252 genes related to energy production, metabolism, stress defense, signaling, protein synthesis and transport and membrane transport. Substantially higher transcript levels were observed for genes encoding proteins like Mg-protoporphyrin IX, chloroplast localized ToxA binding protein (Pr ToxA), CAT, aquaporin Pip1–2, Leu-rich repeat (LRR) receptor kinases, universal stress protein (USP), and isoflavone reductase-like protein 5. Conversely, transgenic plants showed reduced transcript levels of genes encoding malate dehydrogenase, glycogen synthase kinase (GSK), DELLA proteins, and ATP binding cassette (ABC) transporters [113]. To identify transcription factors (TFs) and their downstream genes associated with high CK mediated drought acclimation, transcriptomic profiling of *IPT*-transgenic creeping bentgrass was performed under drought stress. Among 127 differentially expressing TFs, 65 exhibited upregulation and 62 were downregulated in *IPT*-transgenic plants, compared to WT. The downstream genes of 15 TFs also expressed differentially in *IPT*-transgenic plants. Significant transcriptional upregulation was detected in central hubs of *bHLH148*, *MYB4/4*-like and *WRKY28/53/71*. These TFs are ascribed to trigger the genes involved in jasmonic acid (JA) signaling and suppress the genes associated with ABA signaling [114].

Differential proteomic analyses of leaves and roots of *Agrostis stolonifera*, expressing *IPT* under control of two different inducible promoters (SAG12 and HSP18), were performed under heat stress (35 °C). Significant changes were detected in proteins related to energy metabolism, localization and storage, and stress defense. Transgenic plants displayed predominantly higher abundance of enolase, oxygen-evolving enhancer protein 2, putative oxygen-evolving complex, rubisco small subunit, Hsp90, and glycolate oxidase in leaves under heat stress, compared to NT plants. Similarly, root proteome revealed relatively higher abundance of Fd-GOGAT, nucleotide-sugar dehydratase, NAD-dependent isocitrate dehydrogenase, ferredoxin-NADP reductase precursor, putative heterogeneous nuclear ribonucleoprotein A2, ascorbate peroxidase, and dDTP-glucose 4–6-dehydratases-like protein [115]. In another investigation, leaf and root proteome of similarly modified (*pSAG12::IPT*) transgenic creeping bentgrass under drought stress revealed higher abundance of proteins involved in energy production during the processes of photosynthesis and respiration (ribulose 1,5-bisphosphate carboxylase (RuBisCO) and glyceraldehyde phosphate dehydrogenase (GAPDH)), compared to WT plants. Furthermore, transgenic plants showed higher abundance of proteins involved in methionine and glutamine synthesis, chloroplastic elongation factor (EF-Tu), protein disulphide isomerases (PDIs), and antioxidant enzymes (catalase and peroxidase), than WT plants [8]. To elucidate the effects of altered endogenous CK content on the proteome of the chloroplast and its subfractions (stroma and thylakoids), transgenic tobacco plants with high (*pSSU::IPT*) and low (*p35S:CKX1*) endogenous CK were analyzed. Results revealed substantial quantitative differences in stroma proteins of both the transgenic plants but with no qualitative change in chloroplast proteome [116].

## 9. Conclusions

Contrasting observations revealed that CK-deficient plants display a strong stress-tolerant phenotype with increased cell membrane integrity and abscisic acid (ABA) hypersensitivity [26] but at the cost of growth and yield [97]. Alternatively, high CKs in plants facilitate acclimation to osmotic stresses with satisfactory growth and yield by relapsing the conventional transcriptional program activated under abiotic stress [7,58,94,99]. Evaluation of two contrasting Arabidopsis transformants with overexpression of CKX and IPT genes confirmed the constructive role of cytokinins in drought acclimation [55]. CK allows sustainable plant growth and development under stress conditions by stimulating the expression of genes related to growth processes and inhibiting the expression of genes associated with premature senescence. A reciprocal regulation mechanism exists between CK and ABA metabolisms that fine-tunes different processes related stress adaptations as well as plant growth and development. Metabolic resentment between CK and ABA was discovered a long time ago when elevated levels of CK suppressed the activity of xanthine dehydrogenase, one of the key enzymes involved in ABA biosynthesis [117]. Higher concentrations of CK during drought stress act as antagonists to ethylene-induced senescence [118] and biosynthesis, sensitivity, and signaling of ABA (reviewed above). Application of ethylene-based chemical defoliants (thidiazuron and ethephon) upregulated the transcription of CKX and ethylene-related genes in cotton [119], proposing CKX as the common target for CK crosstalk with both ABA and ethylene. Apart from ABA and ethylene, CK crosstalk has also been reported with other phytohormones. *IPT* overexpression caused the upregulation of BR (brassinosteroid)-biosynthesis (DWF5 and HYD1) and BR-signaling (*BRL3*, *BRI1*, *BRH1*, *BIM1*, *SERK1*) genes under water deficit stress, suggesting a positive correlation between CK and BR [9,60]. Conversely, elevated CK concentration repressed JA (jasmonic acid)-related genes (*JAZ12*, *JAZ1*, *OPR2* and *MES3*) [9,60], while high JA concentration in plants attenuated cytokinin signaling by repressing the cytokinin receptor AHK4 and stimulating expression of AHP6, a negative regulator of cytokinin signaling [120]. Contrasting observations were reported while investigating the correlation of IAA with elevated levels of CK. Expression of auxin transport genes, *OsPIN6* and *OsPIN3a*, was downregulated in *IPT* overexpressing rice plants [60] but *pSAG12::IPT* transgenic eggplants displayed relatively higher contents of IAA and GA under drought stress [62]. Nonetheless, more knowledge is required to comprehend interactions among phytohormone signaling pathways. CK mediated drought acclimation has been investigated at length during the past decade but the precise molecular mechanism remains unclear. With the advent of new biotechnological approaches like CRISPR/Cas and the availability of whole genome sequences, future genetic manipulation to enhance abiotic stress tolerance will continuously improve. Future research should focus on excavating novel drought responsive genes/proteins and the molecular mechanisms involved in cytokinin responsive drought tolerance.

## Figures and Tables

**Figure 1 plants-08-00542-f001:**
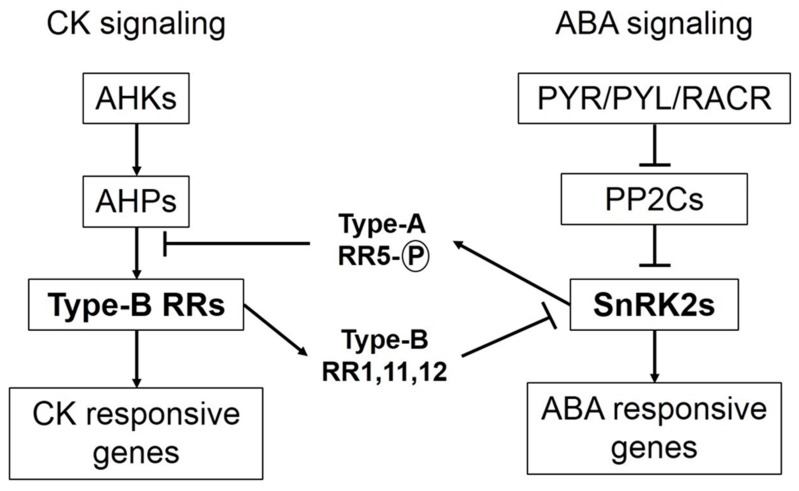
Interaction between signaling components of cytokinin and abscisic acid. Osmotic stresses usually result in enhanced abscisic acid (ABA) biosynthesis and signaling. Binding of ABA receptors to PP2Cs results in activation of SnRK2s, which in turn phosphorylate and activate type-A response regulator 5s (RR5s). Type-A RR5s autoactivate their transcription and act as negative regulators of cytokinin (CK) signaling by hindering phosphate transfer from authentic histidine phosphotransferases (AHPs) to type-B RRs. Conversely, if plants have higher CK concentration (due to exogenous spray of synthetic cytokinins or isopentenyl transferase (IPT) overexpression) under osmotic stresses, CK signaling predominates. Type-B RRs (key components of CK signaling) activate some other type-B RRs (RR1, 11 and 12), which obstruct ABA signaling by inhibiting the activity of SnRK2s.

**Figure 2 plants-08-00542-f002:**
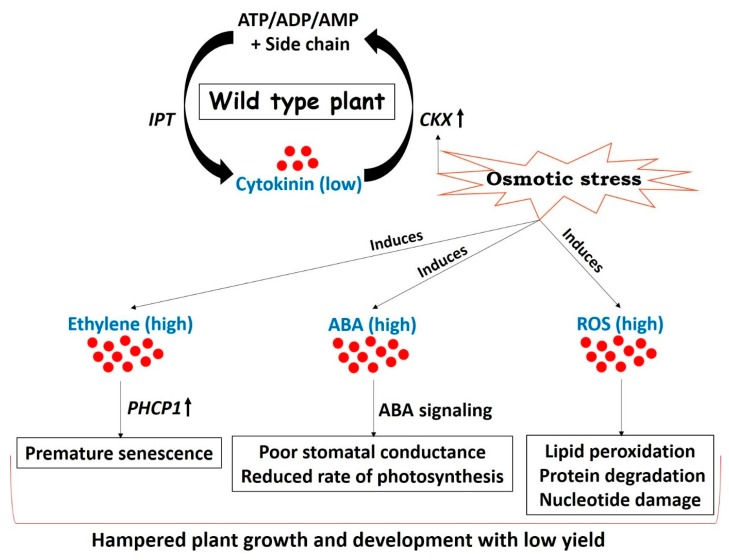
Drought arbitrated adjustments in wild type plants resulting in mild tolerance and low yield. Cytokinin concentration in plants is maintained by two key genes involved in biosynthesis (*IPT*) and degradation (cytokinin oxidase (*CKX*)) of cytokinin. *CKX* is induced under drought stress due to the presence of abiotic stress-inducible cis-elements on its promoter [19,24,25,26], thereby decreasing the concentration of cytokinin. The onset of drought stress enhances the concentrations of stress responsive hormones such as ABA [86] and ethylene [87]. Reactive oxygen species (ROS) are also generated in chloroplasts and mitochondrial electron transport chains in response to stress [88]. Ethylene promotes premature senescence of leaves through a series of coordinated events and enhances the activity of senescence-related Cys protease (PHCP1) [17,18]. Increased biosynthesis and signaling of ABA leads to reduced stomatal conductance and decreases the photosynthetic rate [26,89]. Stress-induced elevated levels of ROS are unfavorable for plant growth and development and are responsible for lipid peroxidation, protein degradation, and nucleotide damage in the worst cases [90,91]. All these stress adaptive events slow down the normal growth and development of a plant resulting in poor yield. (IPT = isopentenyl transferase, CKX = cytokinin oxidase, ABA = abscisic acid, PHCP1 = senescence - related Cys protease, ROS = reactive oxygen species).

**Figure 3 plants-08-00542-f003:**
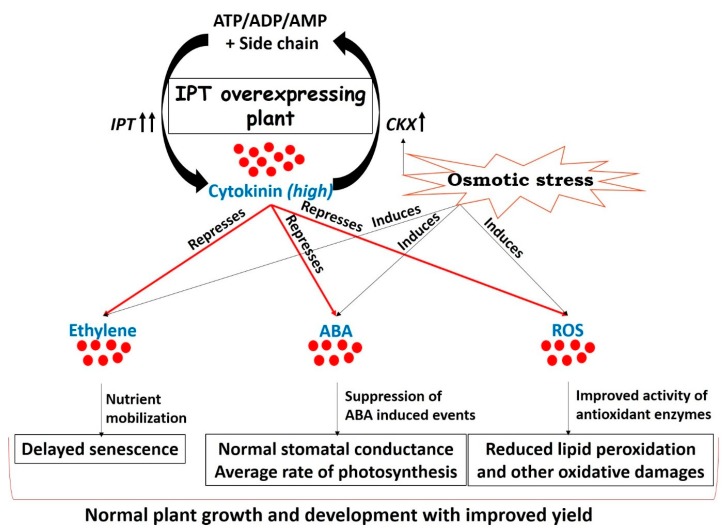
Drought arbitrated adjustments in *IPT* transgenic plants, resulting in tolerance to water deficit stress without compromising yield. Cytokinin concentration in plants is maintained by two key genes involved in biosynthesis (*IPT*) and degradation (*CKX*) of cytokinin. Though *CKX* is induced under drought stress [19,25,26] but overexpression of the bacterial *IPT* gene in transgenic plants helps to maintain cytokinin concentration at higher levels [57,58,59,62] Elevated concentration of cytokinin during drought stress limits the biosynthesis and sensitivity of ethylene [79], helps in nutrient mobilization [8,10], and improves source-sink relations [74]. All these events eventually lead to a delay in the process of senescence in *IPT* overexpressing plants [6,7,58,60,61] or plants sprayed with cytokinin exogenously [53,94]. Increased levels of cytokinin during drought stress also affect biosynthesis [49] and sensitivity [50] of ABA, ultimately reverting the ABA-induced events [4,13]. Hence, *IPT* transgenic plants have better stomatal conductance and improved photosynthetic efficiency [7,58,92] compared to wild types. Elevated cytokinin levels in *IPT* overexpressing plants positively modulate the activities of antioxidant enzymes during drought stress and mitigate ROS driven damages [8,95,96]. Overall, *IPT* transgenic plants sustain normal growth and development under drought stress and offer an improved yield compared to wild types. (IPT = isopentenyl transferase, CKX = cytokinin oxidase, ABA = abscisic acid, ROS = reactive oxygen species).

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
