# Peer review of "The Mode of Cytokinin Functions Assisting Plant Adaptations to Osmotic Stresses"

_plants, 2019, doi:10.3390/plants8120542_

Round 1

Reviewer 1 Report

The manuscript 'Mode of Cytokinin Function assisting Plant  Adaptations to Osmotic Stresses' deals with an important aspect of plant response to osmotic stress, which is mainly secondary stress which can be determined by several abiotic stresses.

The topic is very interesting and very current, considering the climate changes taking place. However, there are some points that deserve a thorough review to improve the article for publication.

In the Introduction the authors seem to focus attention on rice as a cultivated plant that can be subjected to osmotic stress in various ways.  They describe many examples of transgenic rice plants over-expressing IPT that became more tolerant thanks to elevated CK levels. However, there is not a specific conclusion on how to really improve rice tolerance towards osmotic stress in the field.

I suggest to completely rewrite the paragraph of conclusions. At present, it consists of a repetition of concepts repeatedly expressed in the preceding paragraphs with an additional mention of the interactions between the cytokinins pathway and those of other plant hormones. This latter aspect could merit a separate paragraph and a more in-depth discussion.

The conclusion, after a short summary, should help the reader to understand the perspective towards the improvement of osmotic stress tolerance in rice and other crops. The role biotechnology could have for improving resistance to osmotic stress is not mentioned at all.

The authors deal with the mechanism of action of cytokinins bringing mainly examples of transgenic plants or plants treated with cytokinins. The case of mutants that for example exist in Arabidopsis and that helped to better understand the role of cytokinins in the response to abiotic stresses is never discussed.

In conclusion, I suggest to take this suggestion into account and modify the manuscript accordingly.

Reviewer 2 Report

The manuscript deals with the CK and ABA crosstalk related to adaptation to osmotic stress.

According to my opinion, the review is rather descriptive, the mode of action of CK related to osmotic stress is not very well outlined.

Could you please suggest molecular mechanism that is related to CK action that is able to influence plant adaptation/protection against osmotic stress?

On the other hand, the authors deal with a number of works related to crosstalk between ABA and CK in the reaction on osmotic stress conditions.

Could the authors be more exact in the explanation of the crosstalk between ABA and CK? Is there any role played by ethylene at the same time?

Round 2

Reviewer 1 Report

The authors have improved the manuscript according to suggestion, the manuscript is now acceptable for publication.

Reviewer 2 Report

Authors somehow answered given questions and they referred to particular mentions about discussed topics in the article. They also add some information about ethylene, as I asked. According to my opinion, the article can be published in current form.